# fNIRS-Based Upper Limb Motion Intention Recognition Using an Artificial Neural Network for Transhumeral Amputees

**DOI:** 10.3390/s22030726

**Published:** 2022-01-18

**Authors:** Neelum Yousaf Sattar, Zareena Kausar, Syed Ali Usama, Umer Farooq, Muhammad Faizan Shah, Shaheer Muhammad, Razaullah Khan, Mohamed Badran

**Affiliations:** 1Department of Mechatronics and Biomedical Engineering, Air University, Main Campus, PAF Complex, Islamabad 44000, Pakistan; zareena.kausar@mail.au.edu.pk (Z.K.); maill2umer@gmail.com (U.F.); 2Department of Mechanical Engineering, Khwaja Fareed University of Engineering & IT, Rahim Yar Khan 64200, Pakistan; faizan.shah@kfueit.edu.pk; 3Department of Computing, The Hong Kong Polytechnic University, Hung Hom, Hong Kong; postmuhammadshaheer@yahoo.com; 4Institute of Manufacturing, Engineering Management, University of Engineering and Applied Sciences, Swat, Mingora 19060, Pakistan; razaullah@ueas.edu.pk; 5Department of Mechanical Engineering, Faculty of Engineering and Technology, Future University in Egypt, New Cairo 11835, Egypt; mohamed.Badran@fue.edu.eg

**Keywords:** artificial neural network (ANN), functional near-infrared spectroscopy (fNIRS), machine learning, upper-limb prosthesis, transhumeral amputee

## Abstract

Prosthetic arms are designed to assist amputated individuals in the performance of the activities of daily life. Brain machine interfaces are currently employed to enhance the accuracy as well as number of control commands for upper limb prostheses. However, the motion prediction for prosthetic arms and the rehabilitation of amputees suffering from transhumeral amputations is limited. In this paper, functional near-infrared spectroscopy (fNIRS)-based approach for the recognition of human intention for six upper limb motions is proposed. The data were extracted from the study of fifteen healthy subjects and three transhumeral amputees for elbow extension, elbow flexion, wrist pronation, wrist supination, hand open, and hand close. The fNIRS signals were acquired from the motor cortex region of the brain by the commercial NIRSport device. The acquired data samples were filtered using finite impulse response (FIR) filter. Furthermore, signal mean, signal peak and minimum values were computed as feature set. An artificial neural network (ANN) was applied to these data samples. The results show the likelihood of classifying the six arm actions with an accuracy of 78%. The attained results have not yet been reported in any identical study. These achieved fNIRS results for intention detection are promising and suggest that they can be applied for the real-time control of the transhumeral prosthesis.

## 1. Introduction

Amputation refers to the removal of a human limb due to an illness, accident, or trauma. To overcome human limb loss, an artificial device (prosthetics) is provided [1]. The upper limb amputation is divided into five major types, as indicated in Figure 1. Amputees wear transhumeral prosthetic arms to substitute for the loss of elbow and lower portion of arm [2]. A human upper limb can perform seven different motions associated with joints in the arm. Three arm motions are mandatory for transhumeral prosthesis, including elbow extension–flexion, wrist supination–pronation, and hand opening and closing. Advances in the field of biomechatronic have opened new doors to expand the use and applications of prosthetic devices for amputees. However, the control of such prosthetic arms is new area for researchers to explore. Bio signals are preferably used for intention detection that further triggers the implementation of control.

For a long time, upper limb prostheses have been largely controlled using electromyographic (EMG) signals from remnant muscles. Various research studies [4] have considered sEMG for motion intention assessment and used in upper limb prosthetic control. In [5], a DEKA arm with three modular configurations was proposed for people suffering from transradial, transhumeral, and shoulder disarticulations. It utilizes sEMG along with a feet controller and pneumatic bladder for arms control. Lenzi et al. [6] proposed a 5-DoF transhumeral prosthesis for elbow, forearm, wrist, and grasping motions that used an EMG-based low-level controller. Researchers have additionally utilized many other biosensors to control prosthetic arms, such as mechanomyography (MMG) [7], inertial measurement unit (IMU) [8] and near-infrared spectroscopy (NIRS) [9]. Regardless of the above-mentioned developments, a gap exists in the simultaneous control of motions of multi-dimensional transhumeral prostheses.

Signal acquisition and processing are a great challenge in the control of above elbow amputation due to few or no amount of residual muscle and weak muscle activity [4,10,11]. Furthermore, remaining muscle sites for the prosthetic control are not physiologically identified to the distal arm functions [12]. In the past few years, the brain-machine interface (BMI) has appeared as a potential alternative that can offer an incredible opportunity to amputated individuals by empowering them to play out their daily routine [13,14]. It evades the muscles intentions. BMI systems are also implemented to restore motor functions called neuro-prosthesis. The latter assists motor disabled individuals achieve simple everyday tasks [15,16,17]. Quite a few modalities, EEG, MEG, and fMRI, have been considered for BMI applications for their capability to measure brain activities noninvasively. Optical brain imaging has been recently practiced in the BMI field, recognized as functional near-infrared spectroscopy (fNIRS) [18].

fNIRS is useful over the other mentioned modalities for BMI as portability, safety, low noise, and no susceptibility to electrical noise adds to the easy utilization of the system [19]. fNIRS measures hemodynamic response in the cerebral cortical tissue of the brain. The principle of fNIRS uses oxygenated hemoglobin (HbO) and deoxygenated hemoglobin (HbR). The optodes are sensitive to two dissimilar wavelengths in the near-infrared range, 700–1000 nm. This is known as an “optical window”. The biological tissue is somewhat clear to light in this window. The light absorption by water molecules and hemoglobin is relatively low in this region. Henceforth, sensing the light pass through the brain tissue employing noninvasive signal acquisition is performed using an optical source/detector pair, which is placed on the scalp. A relative change in the concentration of HbO and HbR indicates neuronal activation relevant to the executed motion [13]. The attained brain responses relative to distinct motion may comprise of noises that pollute these recorded signals. The noise can be classified as physiological, experimental and instrumental noise [20]. These noises are removed from samples before converting them to magnitude by implementing the modified Beer–Lambert law [21]. The noise recorded because of a computer or neighboring environment is recognized as instrument noise. This noise typically has a high frequency (HF). The HF is separated by applying a low-pass filter. Experimental noise contains motion artefacts, such as head motions or optode dislocation from allotted positions. This generates spikes caused by a variation in light intensities. A study [18] used regularly advanced filtering techniques arbitrarily for noise reduction. Noise is physiologically fashioned [22] as a result of Mayer wave (~0.1 Hz), respiration (0.2~0.5 Hz), and heartbeat (1~1.5 Hz) This is large because of oscillations in blood pressure [23]. One of the chief BMI uses is to extract useful information from raw brain responses for a control–command generation [14]. The captured signals are refined in the four phases: signal preprocessing, feature extraction, classification, and control–command generation. In pre-processing, physiological and instrument artifact and noise are removed. Afterwards the filtration phase, feature extraction step in to gather detailed traits of the signal. Next, the extracted features are classified. The trained classifier is deployed generating control commands using the previously trained data samples [24,25].

Several researchers have embarked on the design and development of robotic arms. The configurations of robotic arms depend on the tasks to be performed by the human arm. The distinct comprehension of actuation approaches is employed. Earlier design approaches have focused on the mechanical issues of structures and the operation of the prosthetic arms [2]. Most of these prosthetic devices are controlled using unnatural methods, such as using the contraction of muscles of the opposite arm [4].

This research attempts to lay a foundation for a framework that offers functionality similar to the human arm, with an intuitive scheme of control. Therefore, by analyzing fNIRS signals to generate control commands for upper limb prosthetic devices, this current study proposes an ANN-based signal classification framework to recognize the intention of six upper limb motions of both healthy and transhumeral amputees. The novelty of the presented research is to generate six control commands using fNIRS for transhumeral amputees. To the best of the authors’ knowledge, there is no existing literature for motion intention detection of six control commands using fNIRS for upper-limb prosthesis applications [1,2,7,14,26].

This paper is organized as follows. Section 2 describes the materials and methods deployed for this study. This includes data acquisition and signal processing. Section 3 consists of feature extraction and classification methods, whereas in Section 4, results are presented and discussed. This includes, filtration, channel selection, and classification accuracies. Section 5 is the conclusion.

## 2. Materials and Methods

In this section, details regarding the experimental procedure followed by the methodology used in signal acquisition and processing, feature extraction, classification, and control command generation are included. A block diagram representation of the methods used is presented in Figure 2.

### 2.1. Subject Information

The study included 15 healthy and three transhumeral amputated subjects among which the healthy subjects were dominant right-hand males. Female amputees are not included simply due to their unavailability. No subject had any psychological, neurological, or optical affliction in the past, as per the recommendation given in [27,28]. The subjects signed a written consent after being briefed about the experimental process. The demographics of transhumeral amputees are given in Table 1. The experiments were allowed by the Human Research Ethics Committee (HREC) of Air University Islamabad. The experiments were performed as per the standards issued by the recent declaration of Helsinki [29].

### 2.2. Optode Placement

The fNIRS data were recorded from the NIRx Imager system, NIRsport (NIRx Medical Technologies, Germany), using an 8 × 8 sensor array positioned on the motor cortex region of the human head scalp. The fNIRS signals were acquired for six arm motions: elbow extension (E.E), elbow flexion (E.F), wrist pronation (W.P), and wrist supination (W.S), hand open (H.O), and hand close (H.C). The optodes were precisely placed on the motor cortex related areas on the 10–20 system that yielded 20 fNIRS channels (10 channels in each hemisphere). Figure 3a shows the position of fNIRS optodes on a healthy subject. Easy cap by NIRx technologies is specially made for optical brain imaging according to international standards [18]. The standard distance between source and detector is 3 cm, as illustrated in Figure 3b [8,22].

### 2.3. Experimental Procedure

The experimental procedure was designed for subjects to perform six motor imagery (MI) tasks. During MI tasks, subjects were instructed to think of performing one of the arm movements and refrain from any other action, such as muscle twitches. The individual subject was asked to perform MI, guided by the experimental team before starting the trials to make them aware of the experimental protocol [29]. During these tasks, the subjects were seated on a comfortable chair to remain relaxed. The chair was placed at an approximate distance of 90 cm from the screen so that the arm motion indications are noticeable and the computer screen backlight does not obstruct the optical sensors [18,30,31].

The experiment session began with an initial rest duration of 30 s to generate a baseline. After that, the routine for the motion was displayed on a computer monitor for subjects to follow. The experiment had two sessions. At first, all tasks were performed in sequence such that the arm motions were pre-defined. However, in the second step, all subjects performed similar arm motions but executed with random intentions. fNIRS logged all six tasks (E.E, E.F, W.P, W.S, H.O, and H.C). Each task/action comprised of ten-second trials with a rest session of 20 s. Each motion was repeated 10 times while in total 12 motions were performed by each subject. An experimental paradigm used in this study is described in Figure 4. A framework of the proposed study is illustrated in Figure 5.

After the signal acquisition, the signals were filtered using FIR filter. These filtered signals were then used to compute the hemodynamic responses using MBLL. Signal mean and peaks were extracted as a feature. The minimum values were also extracted to set the threshold for channel selection. These hemodynamic responses were then fed to the classifying network. Based on the training, the network predicted the motion class. All the details with mathematical equations and numeric values are briefly described in the next section.

### 2.4. Data Acquisition and Processing

This section includes signal acquisition, signal preprocessing, and statistical feature extraction. The signal classification algorithm is also contained within this section.

Before signal recording in fNIRS, an optical imaging technique, the light intensity values were recorded during the oxygenation and deoxygenation of human blood cells of the brain [32]. By using NIRx dual-tip optodes, the light concentration was measured at two values of wavelengths: 760 nm and 850 nm. The acquired light intensities are then processed in nirslab Using this application, one can truncate/remove unwanted data as well as infrequent gaps captured earlier during the acquisition process [2,33]. The dataset can be filtered and hemodynamic states can be computed in the same application as well [34].

#### 2.4.1. Signal Acquisition

The fNIRS signals were acquired using a headset, a flexible cap made of a soft cloth with optodes referred to as EasyCap [35,36]. The experiments were performed with headset placement on the motor cortex in three ways: in the first setting, simply placing the cap; then, the optodes (on the cap) were fixed with spring grommet; and lastly, a complete black cap on top of second set was placed such that the optodes are not visible [35,36,37]. As soon as a subject was wearing the sensor cap, the optodes were calibrated. The results of the first setup are given in Figure 6b, while Figure 6a represents the outcome of the second set.

The rectangles/boxes are the representation of the optodes whether source or detector. The intensity bar on the right is an indication of the optode data quality status. The box changes its color according to the physical status of the sensor to give an idea about which optode to settle on for better signals [38]. The “white color” portrays that there exists no connection between the optodes and the subject scalp. The “red color” indicates a “critical” connection, which directs that it requires the optodes to be adjusted. Commonly, an anomaly is detected because thick hair may be caught in the EasyCap cavity, and just putting the fNIRS optodes again helped to form a better connection. A “yellow color” represents an acceptable connection and the brain signals can be attained. The fNIRS data acquisition system settings were attuned by the system itself. It enhances the gain factor of each optode when the connection is acceptable. The system further saves these numeric values in a “conditions” file. This file can be of help later in the optode selection process. Finally, the “green color” demonstrates that the fNIRS optodes are flawlessly positioned on the subject’s head. This also indicates that an outstanding tie is recognized between the sensing element and the scalp for data acquisition [38].

For the third case, when the data was continuously bad, although connections were fixed by giving a green signal during placement calibration, the dark noise tests were conducted [39]. This test examines the intensity of light that is incident on the optodes from the environment. Keeping in mind, the black covering cap cannot always be worn, dark noise was tested initially with a hypothesis that these special grommets make sure that the least amount of noise is induced to the sensors [40].

#### 2.4.2. Signal Processing

As soon as the optodes were calibrated, the signal acquisition was started [34]. After that, the nirslab software comes with a fNIRS headset to differentiate between bad and good channels based upon the gain values as illustrated in Figure 7. The gain setting allows the exclusion of all channels that have a higher value than a specified value. The nirslab label ‘bad’ to any channel shows that it has a value, at either wavelength, equal to or greater than the threshold value that was specified [34,40]. This value is related to the light intensity of the environment in which the experiment is conducted. Then, the black covering cap was used to address the issue [41].

The discontinuities/spikes caused by the cap placement are removed as illustrated in Figure 8. The number of lines indicates the signal acquired from each optode. Hence, a clean signal can be fed for further processing. The disturbed/noisy and clean signals can be seen in Figure 9 and Figure 10, respectively.

nirslab further provide an artifact removal method.

An acquired hemodynamic response after noise/spike artefact removal from a healthy subject is illustrated in Figure 11.

Additionally, the unwanted or disturbing segments in the signal can be removed according to the values of the threshold or depending on the gain factor incorporated by the machine earlier [8]. A band-pass filter was applied to further smooth the samples to compute hemodynamic states [42]. Filtered data at wavelength 750 nm are illustrated below in Figure 12.

nirslab used “firls” and “filtfilt” MATLAB^®^ instructions for filtering purposes. The “firls” proceeds parameters of linear-segment filtration [8]. The “filtfilt” employs filter parameters to samples. Then, a FIR is introduced. For FIR, a roll-off value states the size of the transition frequency band [34]. The mathematical formation of the filter is given in a set of equations, which are the Fourier transform of the truncated filter and are given in Equations (1) and (2):(1)$$H(\omega )=\frac{1}{2\pi }\int_{-\pi }^{\pi }H_{d}(\lambda )W(\omega -\lambda )d\lambda$$
(2)$$h\left(n\right)=h_{d}\left(n\right)w\left(n\right)$$

The width of transition region between the band pass limits *H* increases with the width of main lobe *W*. It decides the steepness of transition amongst frequencies [43]. This value was by default set to 15 by the nirslab according to signal condition [44]. After filtration, hemodynamic states are computed and are then set to extract features from them.

## 3. Feature Extraction and Classification of Motion Intention Signals

The method of signal feature extraction performs a crucial part in the identification of the discriminatory information carried by the bio signals [32,44,45,46]. This section details the features extracted from the dataset and the details of the applied machine learning algorithm for motion classifications.

### 3.1. Feature Extraction

To execute control commands for six arm motions, features for the signal classification were extracted. For fNIRS brain signals, signal means (SM), signal peak (SP), and signal minimum (min) [43,47] were extracted for thresholding purposes. The signal mean was calculated as (3):(3)$$\mathrm{SM}=\frac{1}{N}\sum_{i=1}^{N}X_{i}$$
where N represents the total data points and *X_i_* represents the signal amplitude value. The signal peak was calculated using a signal amplitude variation between two head-to-head sections that exceeds a pre-defined threshold value to cut noise. It is given by (4):(4)$$\mathrm{SP}=\sum_{i=1}^{N}f\left(\left|X_{i}-{\;X}_{i+1}\right|\right)$$

As per the existing literature [38], SM and SP offer improved control performance for fNIRS-based systems. However, regarding the stated possibility of an initial fNIRS signal dip, a (min) signal value was added as a feature [43]. The features were calculated from only selected optodes based on criteria described earlier in this section using a 2 s moving window. MATLAB^®^ was employed to execute all the features computation.

### 3.2. Artificial Neural Network (ANN)

For the evaluation of the performance of acquired fNIRS signals, a widely used [38] classifier in pattern recognition was implemented, namely, artificial neural network (ANN). It uses various neuron layers to plan information starting with one circulation then onto the next one for better and enhanced results, i.e., returning less error [46,48]. A system called backpropagation assists ANN to form a bridge between input and output layers in which the corresponding labels/indicators are present [49]. The machine learning toolbox designed by MATLAB^®^ for neural networks came into play for training the samples [50]. When using the toolbox, all you need to set is the number of hidden neurons in the layer of this artificial neural network [51]. The designed model then estimates the error of the probable output in contrast with the actual output. The network further explores the error to variate to adjust the weightage, to minimize the generated error for the next cycle, and this process continues unless the error approaches zero [46]. For the network, the rule activation function was applied, and the weights were randomly assigned by the toolbox. The results and ANN training particulars are presented in the next section. The comprehensive flow diagram of the ANN classifier is shown in Figure 13.

The ANN network had 2 hidden layers with 12 and 6 neurons. The output layer will give one definitive class, i.e., motion class, as defined in Figure 14. First, preprocessed information is passed through the first layers, which contain 128 filters with a kernel size of 12. The output from the first layer is 24 × 128 [52]. The second layer contains the same number of filters and a kernel size of six. The output from the second layer is 12 × 128. Subsequently, the global average pooling is applied between the output layer for which the Adam optimization method was deployed [53,54].

After classifying the fNIRS signals, the trained model was tested. The obtained results are discussed in the next section.

## 4. Results and Discussion

In this study, fNIRS signals were acquired to generate control commands of human arm motions for the transhumeral amputee. The fNIRS hemodynamic responses are acquired employing optical sensors, i.e., optodes. These responses are used to drive a prosthetic arm for transhumeral amputees. For optical sensors, dark noise plays an important role. The NIRx Technologies designed a high-functioning device that can be used for real-time purpose. However, the difficulties have never been addressed before. For this study, the results were not only analyzed based upon the accuracy of the motion classification, but also the sustainability for real-time applications. The optode placement problem was addressed and researched upon and is detailed in previous sections. This section specifically presents the experimental results achieved using the proposed framework. Window sizing of various lengths and durations has been observed in the literature to detect fNIRS signals [31]. The period of 0–0.5 and 0–1 s was selected [55,56]. This split time window was used to inspect hemodynamic response. This then determines the optimal window, which then generates a command with a minimum amount of time.

### 4.1. Channel Selection

The electrical gain component that was adjusted to the absorption spectra is shown by number 6 in Figure 7. The photocurrent generated by the optical wave is amplified to a greater extent since this factor rises in value [41]. As the gain component increases, the signal-to-noise ratio of the input drops [42]. As a result, nirslab may identify channels with gain factors greater than a preset value and reject them from further processing and analysis. The ordered pair (1.8477 1.7928) reflects the values of a metric we employed to quantify the raw data’s signal-to-noise ratio [45,46]. The coefficient of variation (CV) is the measure. Since there are different measuring wavelengths, two CV values are reported. The sampling frequency is 78.1 Hz. After the analysis presented earlier, a framework is presented here to overcome the issues contributing to the bad and unsteady signals from the fNIRS sensor [8,34].

### 4.2. Motion Classification Accuracy

The classification results were then analysed subject-wise and then the average accuracy of all subjects was calculated. The difference in the classification accuracy is evidence of the importance of signal acquisition and processing procedure. The reliability of the results is very well dependant on how the signals were acquired and further processed [8,47]. A noisy and disturbing signal resulted in error, while the refined signals were easy to handle by the classifier algorithms and hence a minimum error was generated [47,52,57]. Furthermore, a *t*-test was applied to student participants. This test checks the statistical significance of the attained results [54]. The confidence interval is specified at 95% (*p* < 0.05). A quantifiable comparison between healthy subjects and amputees was not possible due to a restricted number of amputees. However, the computed *p*-value is 0.0248, with a 95% confidence interval within healthy subjects.

The network used a sigmoid function for gradient descent. A total of 60% of the samples were fed for training and 20% samples were utilized by the network for testing and validation each. Each step was calculated in 320 µs with a 3 s epoch completion time. Then, a confusion matrix was extracted as soon as the training ends, which not only indicates the number of samples, those that were accurately classified, but also the false samples that generated an error. The number of hidden neurons was set to 20. Twelve neurons were present in each of the transitional hidden layers. The neuron number in the output layer is specified as six.

The healthy subject-wise accuracy is illustrated in Table 2, whereas Table 3 represents the accuracy of amputed subjects.

As stated in Section 1, no work has been conducted for transhumeral amputees in non-invasive manner to generate six number of control commands. However, the studies that used fNIRS for some other applications are presented below in tabular form for accuracy and number of control commands comparison. The study comparison is illustrated in Table 4.

It can be seen from the studies above that, as the number of control commands increase, the accuracy decrease. However, the time response of classifiers has no trend. It is due to the fact that these studies have been conducted on the signal set acquired by third parties via online forums. In the proposed framework, the signals were acquired and then analyzed. Based on the conditions during the signal acquisition process, further steps were taken, such as filtration and channel selection. This makes a difference, as documented by [22,58]. The results show the potential usability of the presented framework in real-time applications and is a step towards enhanced motion prediction in BMI applications.

## 5. Conclusions

In this research study, an fNIRS-based approach was investigated to recognize the motion intention of the human upper limb. The fNIRS signals are acquired from the motor cortex region of the brain using NIRSport from NIRx Technology. The fNIRS signals were acquired for six arm motions. These motions included elbow extension (E.E), elbow flexion (E.F), wrist supination (W.S), wrist pronation (W.P), hand open (H.O), and hand close (H.C). Channel selection was conducted based on the gain values computed during signal acquisition. An FIR filter was applied to filter the samples. Signal mean, signal peak and minimum value were computed as a feature set. ANN classifier was trained for motion intention prediction. On average, the motion intention prediction was 78% (*p* < 0.05) and 64% accurate for healthy and amputated subjects, respectively. The highest accuracy for an individual subject was recorded as 79.6%. A possible extension of the presented work includes the framework design for accuracy enhancement and eliminating the channel selection complications. The application of the presented approach with the increased number of arm motions, incorporating individuals of different age groups, and the implementation of generated control commands to control a prosthetic arm device in a real-time setting are some other directions for future work.

## Figures and Tables

**Figure 1 sensors-22-00726-f001:**
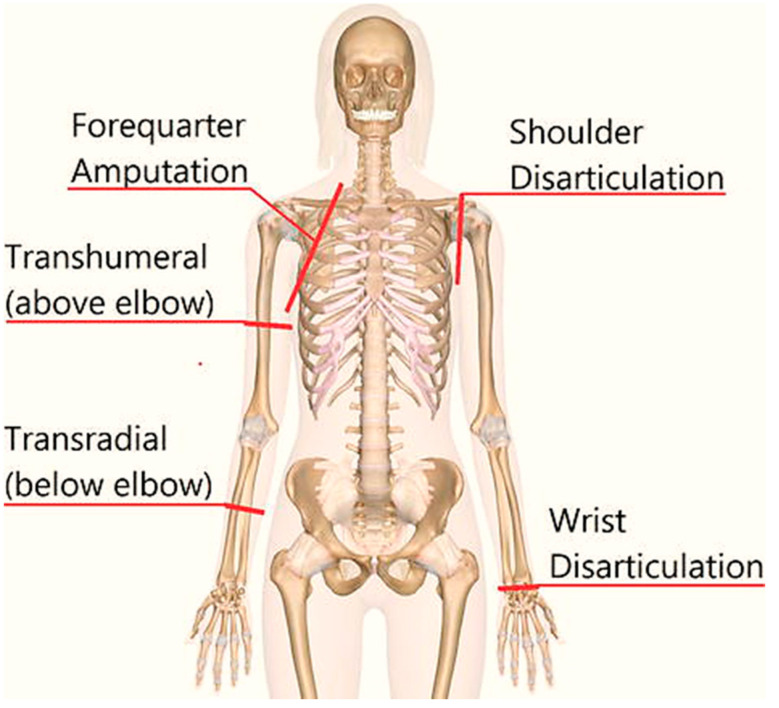
Levels of upper limb amputation [3].

**Figure 2 sensors-22-00726-f002:**
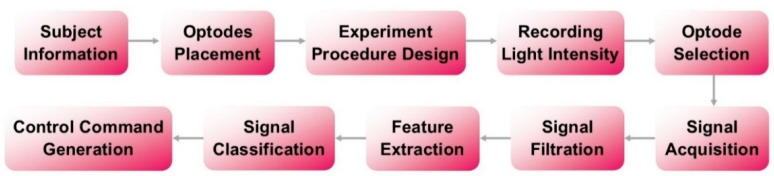
Methodology of the study.

**Figure 3 sensors-22-00726-f003:**
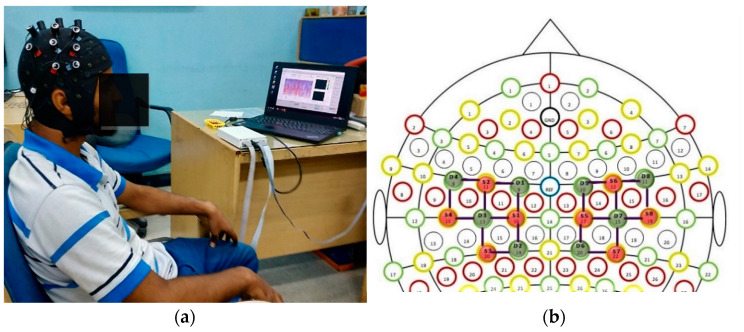
Signal acquisition environment and optode placement. (**a**) Experiment setup showing optode placement on the motor cortex of a healthy subject. (**b**) Eight sources (S) and eight detectors (D) were positioned on the subject’s motor cortex region of the brain to record fNIRS signals with a separation of 3 cm resulting in twenty channels.

**Figure 4 sensors-22-00726-f004:**
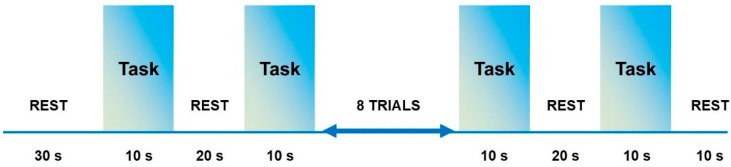
After an initial 30 s rest, each functional near-infrared spectroscopy block consists of 10 s activations and 20 s rests. The total experiment duration for acquiring fNIRS signals is 11 min and includes 12 trials in total.

**Figure 5 sensors-22-00726-f005:**
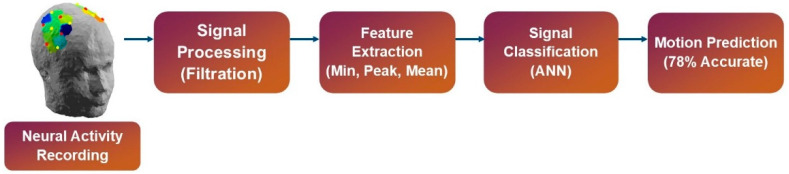
Flow diagram of fNIRS-based motion intention recognition for the transhumeral prosthesis.

**Figure 6 sensors-22-00726-f006:**
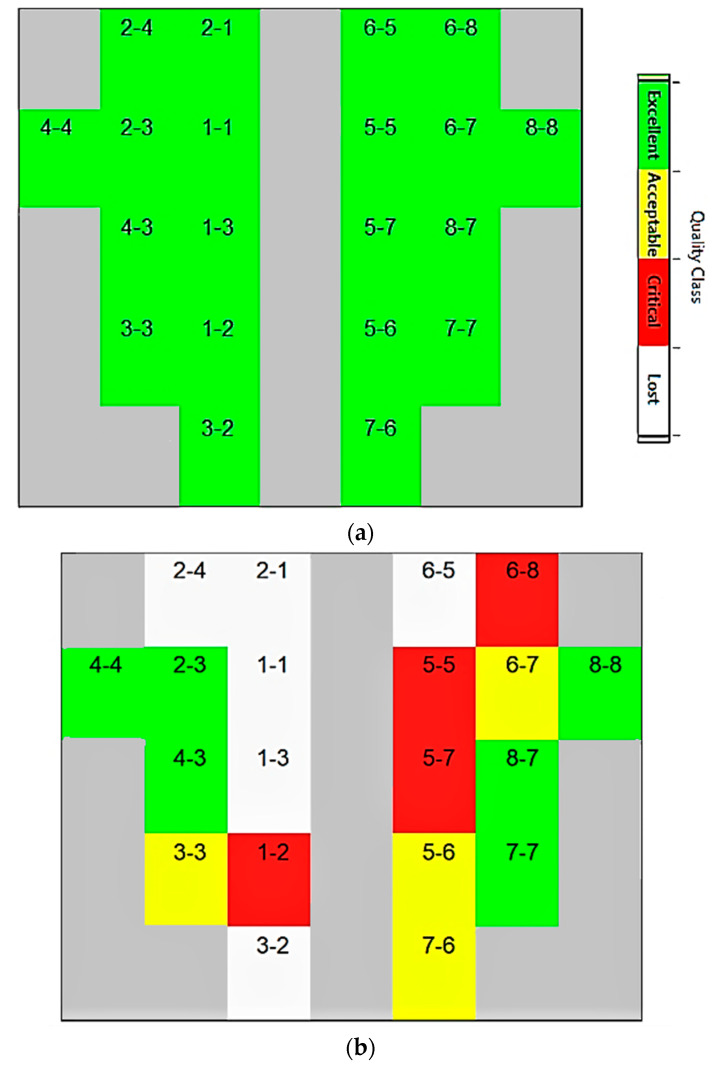
Optode status window. (**a**) Flawless optode connection with head scalp; (**b**) faulty optode connection. The signal quality class can be read from a color bar shown along with the optode status window.

**Figure 7 sensors-22-00726-f007:**
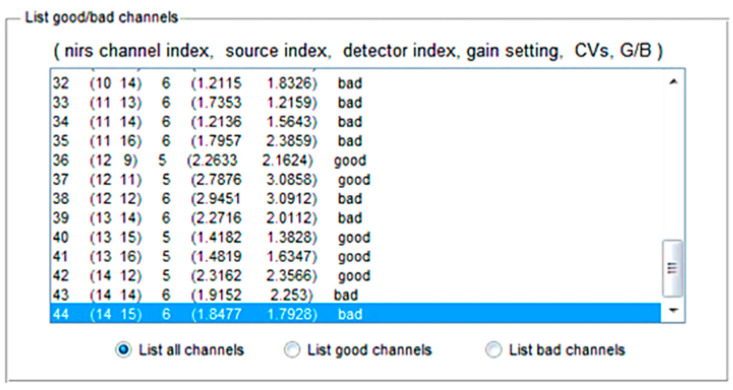
List of good/bad channels to remove bad channels from the analysis and signal classification.

**Figure 8 sensors-22-00726-f008:**
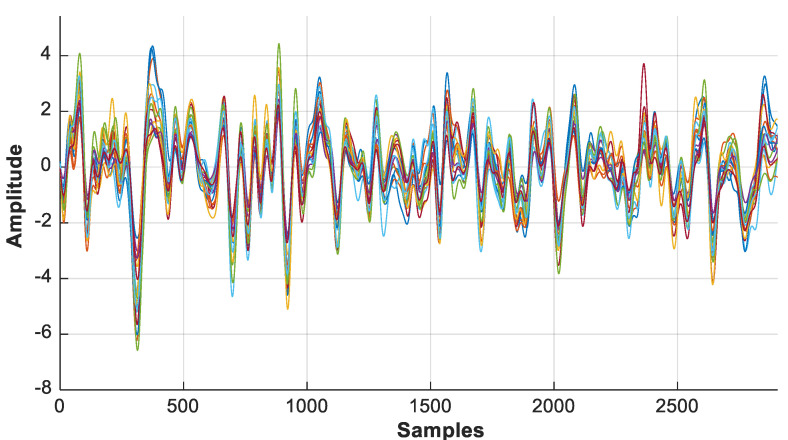
Visualization of recorded raw light intensity of each optode.

**Figure 9 sensors-22-00726-f009:**
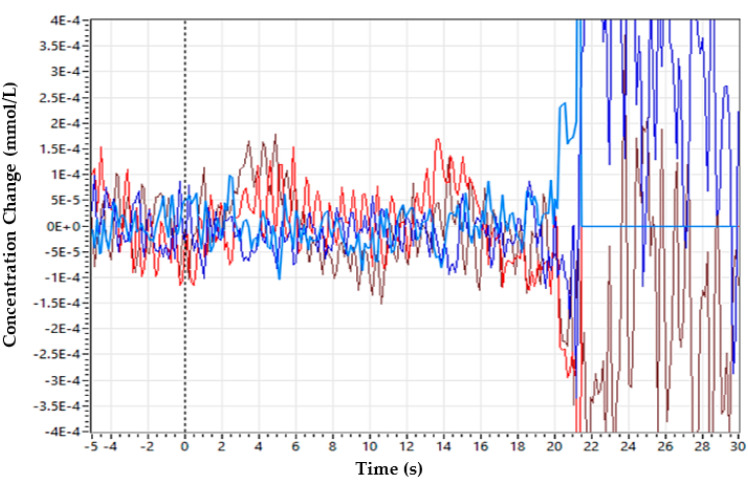
Disturbed/noisy signal before grommets and covering head cap is incorporated. Blue lines represent the data acquired from the detector while the red line represents the source signal.

**Figure 10 sensors-22-00726-f010:**
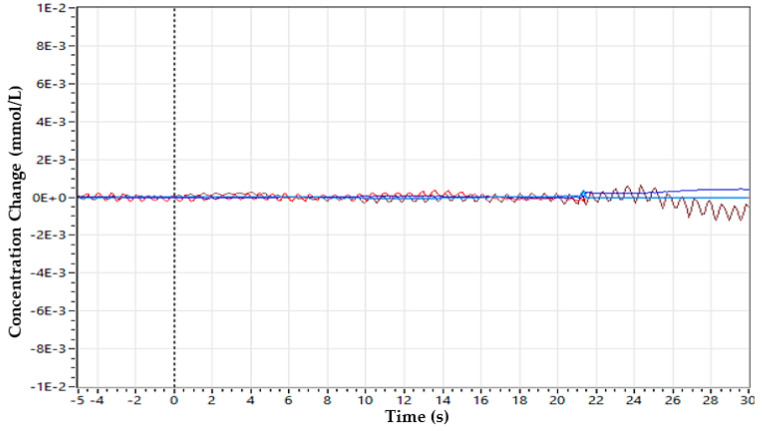
Clean signal after incorporation of grommets incorporated with covering head cap.

**Figure 11 sensors-22-00726-f011:**
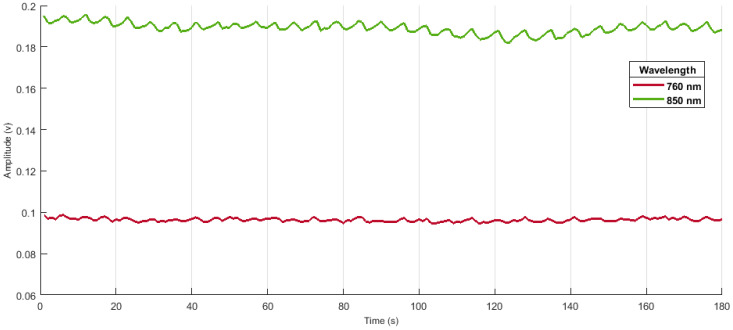
The fNIRS data obtained from a healthy subject according to the experimental protocol. The upper signal is from the source of 760 nm, and the lower one represents the 850 nm wavelength.

**Figure 12 sensors-22-00726-f012:**
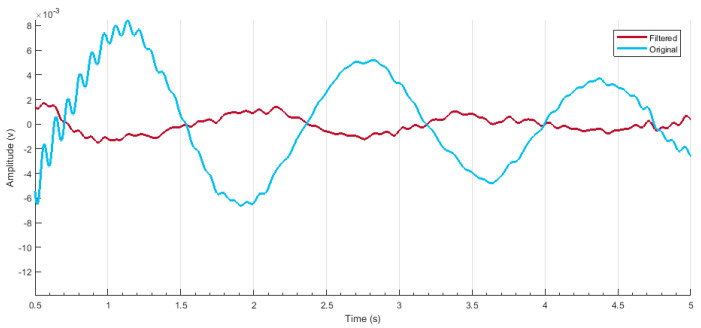
The filtered sample after the implementation of band-pass filter in a range of 0.01 Hz–0.2 Hz.

**Figure 13 sensors-22-00726-f013:**
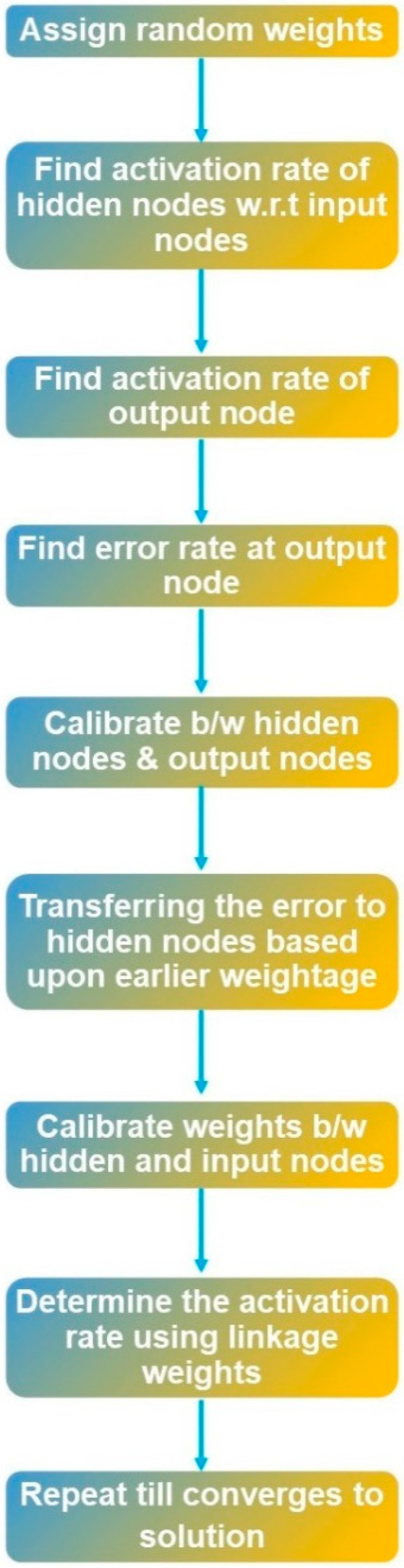
Flow diagram of the ANN classifier.

**Figure 14 sensors-22-00726-f014:**
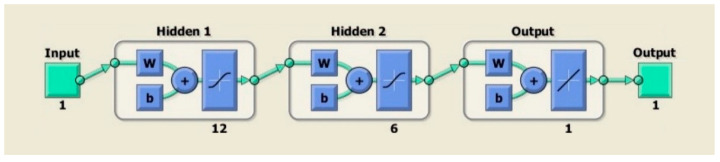
ANN network architecture.

**Table 1 sensors-22-00726-t001:** Demographic characteristics of amputee subjects.

Amputee Title	A1	A2	A3
Gender	Male	Male	Male
Age	23	32	42
Amputated Side	Right	Left	Right
Residual Limb Length	14 cm	18 cm	10 cm
Time since Amputation	7 Months	24 Months	145 Months

**Table 2 sensors-22-00726-t002:** Offline classification accuracies of fifteen healthy subjects using single features signal mean (SM), signal peak (SP), signal minimum (SMin) and waveform length features for EMG and fNIRS using LDA and ANN Classifiers.

Features	S1	S2	S3	S4	S5
**SM**	72.88	61.1	74.89	68.63	69.63
**SP**	67.85	68.89	76.9	69.03	70.03
**SMin**	74.99	63.84	73.49	68.13	69.13
	S6	S7	S8	S9	S10
**SM**	75.04	75.38	77.15	67.86	74.56
**SP**	72.22	72.28	65.38	75.77	69.76
**SMin**	77	74.44	71.15	71.75	66.62
	S11	S12	S13	S14	S15
**SM**	66.78	64.76	60.37	71.54	79.6
**SP**	69.74	69.56	65.47	71.94	59.81
**SMin**	66.87	70.61	64.74	69.22	67.6

**Table 3 sensors-22-00726-t003:** Offline classification accuracies of three amputee subjects using single features signal mean (SM), signal peak (SP), signal minimum (SMin), and waveform length features for EMG and fNIRS using LDA and ANN Classifiers.

Features	A1	A2	A3
**SM**	69.26	61.65	57.05
**SP**	68.91	60.18	57.72
**SMin**	55.1	50.05	51.93

**Table 4 sensors-22-00726-t004:** Performance evaluation and comparison with existing classification models.

Technique	Learning Method	Time Response	Number of Control Commands	Classification Accuracy
TD features [58]	LDA	5.5 s	2	72.82%
FD features [59]	LDA/SVM	15 s	2	83%
Raw fNIRS [22]	ANN	4 s	4	58%
TD features [60]	SVM	0.5 s	6	68.1%
Proposed framework	ANN	320 µs	6	78.65%

## Data Availability

It is a funded project and hence data is not publicly available. However, the data can be made available upon request.

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
