# Peer review of "fNIRS-Based Upper Limb Motion Intention Recognition Using an Artificial Neural Network for Transhumeral Amputees"

_sensors, 2022, doi:10.3390/s22030726_

Round 1

Reviewer 1 Report

In general, the research area is relevant and as such quite a bit explored by other scientists. There are quite numerous papers where this specific approach was presented (signal acquisition->filtration->features extraction->using ANN to identify certain patterns in signals). Using the functional near-infrared spectroscopy (fNIRS)-based approach for recognition of human intention(s) is very interesting and not easy to find in other publications so this gives this paper certain level of novelty making it worth to be consider for publication.

The research looks coherent, tools and methods seem well thought and properly selected (e.g. electrodes placement is correct and focusing on the "plan & execute" brain area, using FIR band-pass filter also is correct), The results confirm that the solution guarantees a very good level of accuracy.

However, I have a few reservations over the presentation:

  1. Authors mentioned using FIR band-pass filter. However, the filter parameters are nowhere to be found in the paper. Also, it would make a lot of sense to provide some explanation how these parameters were determined (I assume that the Authors have spent some time on the filter design so not including any discussion in this regard is quite an omission.
  2. There is a generic description of the ANN and some brief outline of its architecture (number of layers, number of neurons per layer, etc.). However, I would like to know (and I believe I am not the only one) more about the training process: what was the training data (as they are used as a pattern), where did it come from, how was it validated (which is essential - only data that is unquestionably appropriate guarantee correct network performance), etc. Such an information would be much more interesting than e.g. somewhat trivial Figure 13 which presents a very typical ANN training process.

Other than that I think the paper quality does not leave much to be desired. Language does not require any substantial improvements. Hence, after improving the paper in the two above areas I would recommend this paper for publication.

Author Response

The authors would like to thank the reviewer for his/her valuable comments. We have tried our best to remove reviewer concerns in the revised manuscript all revisions are made in the yellow highlighted text in the updated version of the manuscript.

Reviewer 2 Report

The manuscript needs English proofreading.

Author Response

Thank you for your concern. The manuscript has been revised as desired.

Reviewer 3 Report

  1. Line 36 - exclusion or maybe removal?
  2. Paper full of grammar errors, such as e.g. line 50 - it is not a present continuos. 
  3. Why only 3 patients with amputated limbs participate in the study. What was the reason for not involving females? Are you planning to expand this study?
  4. Were all patients (those with amputations) rich- or left- handed? (dominant)?
  5. Did you compare your results with other similar?

Author Response

The authors would like to thank the reviewer for his/her valuable comments. We have tried our best to remove reviewer concerns in the revised manuscript. All revisions are made in the yellow-highlighted text in the updated version of the manuscript.

Reviewer 4 Report

This article presents the design ANN based on fNIRS for motion recognition in amputees. 

General Judgment Comments

Overall, the article is overall good and well written, although some details may be beyond the scope of the article (e.g. details about the software used may be redundant as they are available in the nirslab manual). Additionally, there seems to be an extra set of recordings  (EMG) not reported in the methods. More details should instead be reported about the methods and results. I have detailed my comments below.  

Suggestion: major revision 

The article is overall of sufficient quality. I recommend the editor to accept the manuscript given that revisions are made.

Major Issues

  • The method section is well detailed, however, some information may be redundant with the nirx manual. Why I understand the importance of reporting how bad and good channels are identified, I think knowing how many were labeled as usable would be a piece of more meaningful information to evaluate the current set of results. I would suggest the authors to reduce the amount of information from the nirx manual, and instead reporting more details about the current work (e.g. number of channels excluded, total number of signals fed to the ANN, etc).
  • Some figures are hard to compare and add little to none to the manuscript. For example, Figures 9 and 10 are on two different y-axis scales, making the comparison difficult, and moreover, they add little to the paper. I would suggest the authors to remove these figures, as well as the figures from 6 to 12 included. 
  • Would you decide to keep figure 6, what is the second set mentioned on Line 205? 
  • Some of the abbreviations are used but never introduced, such as sEMG should be written down in its extended version on line 52, while the abbreviation for the Beer-Lamber Law should be added in parenthesis in Line 88. I would suggest the authors to go through the paper and check whether the different abbreviations have been introduced properly. 
  • Whas it the EMG indicated in Table 2 and 3? Does this means you have combined EMG and fNIRS measures? If so, detailes about the EMG recordings should be added in the methods. 

Minor Issues

  • A comma is missing in the abstract on Line 25.
  • For reproducibility purposes, please add the nirslab version and platform (operating system) on Line 195.
  • There is either a missing comma or incomplete sentence on Line 363
  • There are two extra commas on Line 386. 

Final comments

The article is overall good and presents an interesting project. Major edits are suggested before the article is accepted for publication.

Author Response

(The authors gave the same response as above.)

Round 2

Reviewer 4 Report

Thank you for the revisions